# Electrical Characterization and Simulation of GaN-on-Si Pseudo-Vertical MOSFETs with Frequency-Dependent Gate C–V Investigation

**DOI:** 10.3390/mi16111193

**Published:** 2025-10-22

**Authors:** Valentin Ackermann, Mohammed El Amrani, Blend Mohamad, Riadh Ben Abbes, Matthew Charles, Sebastien Cavalaglio, Manuel Manrique, Julien Buckley, Bassem Salem

**Affiliations:** 1Univ. Grenoble Alpes, CEA, Leti, F-38000 Grenoble, France; mohammed.elamrani@cea.fr (M.E.A.); blend.mohamad@cea.fr (B.M.); matthew.charles@cea.fr (M.C.); julien.buckley@cea.fr (J.B.); 2Univ. Grenoble Alpes, CNRS, CEA/Leti Minatec, Grenoble INP, LTM, F-38054 Grenoble, France; riadh.benabbes@cea.fr (R.B.A.); sebastien.cavalaglio@cea.fr (S.C.); manuel.manrique@cea.fr (M.M.); 3GREMAN UMR 7347, Université de Tours, CNRS, INSA Centre Val de Loire, 37071 Tours, France

**Keywords:** GaN-on-Si, MOSFET, pseudo-vertical device, device processing, electrical characterization, TCAD simulations, traps defects, gate morphology, RC effect

## Abstract

This work presents a comprehensive study of GaN-on-Si pseudo-vertical MOSFETs focusing on single-trench and multi-trench designs. Thanks to a gate-first process flow based on an Al_2_O_3_/TiN MOS stack, both fabricated devices exhibit promising transistor behavior, with steady normally OFF operation, very low gate leakage current, and good switching performance. Following the extraction of a low effective channel mobility, the frequency dependence of gate-to-source C–V characteristics is studied. In addition, using TCAD Sentaurus Synopsys simulations, the impact of positive fixed charge and donor-type defects at the p-GaN/dielectric interface is investigated, together with the frequency dependency. Finally, by comparing experimental and simulated results, a mechanism is proposed linking the observed threshold voltage shift to the presence of sharp trench-bottom micro-trenching. This mechanism may further explain the origin of the additional C–V hump observed at high frequencies, which could arise from charge trapping at the p-GaN/dielectric interface or from charge inversion in the p-GaN region.

## 1. Introduction

Wide-bandgap semiconductors are progressively transforming the landscape of power electronics, with gallium nitride (GaN) emerging as a prime candidate for high-power efficiency and high-voltage devices. Owing to its high critical electric field and high electron mobility, GaN could enable the development of power transistors with high breakdown, low specific on-resistance, and fast switching speeds [1]. In particular, vertical architectures are advantageous for medium- to high-voltage applications, as they offer improved scalability, better current spreading through the device, reduced chip area, and higher breakdown voltage compared to their lateral counterparts [2]. Most importantly, vertical devices could have avalanche capability, which is harder to achieve in lateral hetero-junction devices.

The majority of vertical GaN transistors discussed in the literature have been processed on low-defect density and expensive bulk GaN substrates, which are only available in limited sizes [3,4,5,6,7]. However, a growing number of research groups are beginning to successfully integrate vertical GaN devices on more mature silicon substrates [8,9,10]. Indeed, the GaN-on-Si platform has matured considerably so far, as this promises the performance benefits of GaN with the cost advantages of large-diameter, scalable, and low-cost Si wafers.

In this context, the vertical trench-MOSFET remains a compelling solution due to the benefits of the MOS gate technology and compatibility with gate drive circuits commonly employed in silicon power technologies. However, achieving high-performance GaN-on-Si vertical MOSFETs requires precise control over material interfaces (particularly the MOS gate) [10,11,12], trench morphology [4,13], and dielectric reliability [14,15].

Despite the potential of GaN-on-Si vertical MOSFETs, challenges still persist at the gate level that can compromise device performance and reliability. A major concern is the presence of charge trapping and interface defects at the p-GaN/oxide interface, which can lead to V_th_ instabilities and degraded switching behavior [16,17]. These effects are particularly pronounced under high-electric-field or prolonged gate-bias conditions. In addition, the trench morphology is critical, as the etching processes can lead to surface damage and roughness at both the trench sidewall and trench bottom, which degrade the quality of the gate dielectric/GaN interface. Furthermore, the trench geometry inherently enhances electric field crowding, particularly at sharp corners, leading to intensified stress on the dielectric, and consequently, increasing trapping effects [13,18]. These combined effects can lead to poor electron mobility all around the gate, threshold voltage instability, increased gate leakage, and reduced long-term reliability. Addressing these gate-related limitations is essential for improving the performance and reliability of GaN-on-Si vertical MOSFETs and forms the basis of this study.

This work is divided into two parts. First, we present a gate-first process to fabricate two GaN-on-Si vertical MOSFET architectures: one based on a single-trench layout, and another using a multi-trench design. Using low-voltage I–V characterization, we compare the transistor behavior for the two architectures and extract a low effective channel mobility. Secondly, this channel mobility is investigated through gate-to-source capacitance–voltage measurements (C-V) at multiple frequencies, and different explanations are examined by TCAD simulations.

## 2. Materials and Methods

The GaN layers were epitaxially grown on a 200 mm silicon (111) wafer using metal–organic vapor phase epitaxy (MOVPE). First, buffer layers of AlN and AlGaN were grown on the silicon wafer; then, from bottom to top, the different doped GaN layers were grown as follows: 500 nm n^+^ GaN drain layer (Si, 5 × 10^18^ cm^−3^); 1 µm n^-^ GaN drift layer (Si, 2 × 10^16^ cm^−3^); 500 nm p-GaN (Mg, 5 × 10^18^ cm^−3^); and 100 nm n^+^ GaN source layer (Si, 5 × 10^18^ cm^−3^). The wafer was diced into 2 × 2 cm squares which were then used for device fabrication.

The process flow started with the deposition of a 300 nm SiO_2_ hard mask by PECVD. The GaN gate trench and mesa structures that terminate the n-p-n heterostructure were then formed using CF_4_ dry-etching in an inductive couple plasma reactor (ICP-RIE) to etch the SiO_2_ followed by ICP-RIE using a Cl_2_-based dry-etch process to etch the GaN ~700 nm through the n-p-n epi-layers. After etching, a 10 min rapid thermal anneal (RTA) at 600 °C in O_2_ atmosphere was performed to activate the Mg in the p-GaN layer.

Following the trench fabrication, a HCl pre-deposition wet surface treatment was applied to the sample for 5 min at an ambient temperature. Then, 40 nm of thermal Al_2_O_3_ was deposited by ALD at 300 °C using a trimethylaluminium (TMA) and H_2_O vapor for the deposition, followed by a sputtered deposition of 40 nm of TiN as the gate metal. The metal gate was then etched to reveal the source and drain areas using a SF_6_/O_2_-based dry-etch, followed by the removal of the Al_2_O_3_ layer using a BCl_3_-based dry-etch. The gate fabrication was completed by a 4 min RTA at 400 °C in N_2_ atmosphere.

Next, the drain area was formed using the same Cl_2_ dry-etch as above, etching through the n^-^ GaN drift layer to the n^+^ GaN drain layer, before e-beam deposition of the Ti/Au drain contacts. Finally, the source contacts made of the same metal stack were deposited, after opening the SiO_2_ hardmask using a dry-etch fluorocarbide process.

Figure 1 shows a cross-section schematic of a processed pseudo-vertical single-trench MOSFET, along with its top-view SEM image. Additionally, large-area MOSFETs based-on a multi-trench design were also fabricated during the process fabrication, as illustrated in Figure 1b. As can be seen, the resulting MOS gate is located inside the trench as well as all around the mesas for both structures. Depending on the device studied, the dimension of the expected gate trench width (W_tr_) is 3 µm for the single-trench MOSFET and 9 µm for the multi-trench MOSFET.

The bright-field STEM image in Figure 2 shows the trench gate of a single-trench MOSFET. While the deposited dielectric and gate metal show conformal coverage of the GaN surface, the STEM image also shows sloped trench sidewalls and micro-trenching at the trench bottom corners, the result of our trench-etching process. These trench features, particularly that of the bottom corners, could lead to high-electric-field crowding around these areas, potentially weakening the gate reliability.

## 3. Results and Discussion

Figure 3 illustrates the forward (a), (b) and forward–backward (c) transfer characteristics of a single-trench device. These show normally OFF transistor behavior, with a threshold voltage (V_th, lin_) of 2.5 V, extracted through extrapolation of the linear region of the characteristic at V_D_ = 3 V. The threshold voltage is relatively stable, with a V_th_ hysteresis of 300 mV, as shown in Figure 3c. The device demonstrates a high I_ON_/I_OFF_ current ratio of 10^8^ and a subthreshold slope of 347 mV/dec. In addition, the gate has very low gate leakage current (I_G_ ~ 100 pA) as observed on the inset of Figure 3b, due to the 40 nm gate dielectric [19].

The normalized gate-to-source C-V characteristic at 200 Hz of a single-trench MOSFET is shown on Figure 4. The V_th, C-V_ extracted at the midpoint between the minimum capacitance and the gate oxide capacitance (C_ox_) is comparable to that extracted from the semi-log transfer characteristics (V_th, log_), due to the more static regime related to the low frequency of the C-V measurement.

Using the gate oxide capacitance from Figure 4 together with the transconductance from Figure 3a, one can extract the channel field-effect mobility, defined as follows:(1)$$\mu_{\mathrm{ch}}={\;g}_{m}\cdot \frac{L}{Z}\cdot \frac{1}{C_{\mathrm{ox}}}\cdot \frac{1}{V_{D}}$$
where g_m_ is the transconductance, Z is the channel width of 685 µm, L is the channel length of 0.62 µm, C_ox_ is the gate oxide capacitance 166 nF/cm^2^, and V_D_ is the drain voltage defined at 0.1 V. Consequently, Figure 5 represents the field-effect mobility estimated at this low drain voltage. A channel mobility of 7.2 cm^2^/V·s is extracted, whose low value could be explained by carrier scattering coming from surface roughness on the p-GaN sidewalls [4] and/or by a high density of oxide interface traps at the dielectric/GaN interface [20,21].

Figure 6 shows the forward transfer characteristic of a multi-trench device composed of 159 trenches (N_tr_). The transfer characteristics demonstrates a normally OFF behavior of the transistor with a V_th, lin_ of 1.2 V at V_D_ = 3 V, and switching capabilities with a I_ON_/I_OFF_ current ratio of 10^5^ and a subthreshold slope of 282 mV/dec. The drain current is multiplied by a factor of 2× to a higher number of trenches in our device, while the gate leakage is still similar to the single-trench device. The fact that there is a slight difference in the device switching (V_th_, I_OFF_ …) between the two architectures could be explained by the wider device area and the complex multi-trench design. This design could be more subjected to process non-uniformities on the gate module (p-GaN doping, GaN/dielectric interface and bulk dielectric quality), leading to potential trenches with weaker switching properties affecting the whole device.

Low channel mobility is investigated by gate-to-source C-V measurements at a wide range of frequencies, as shown in Figure 7. Increasing the frequency to 10 kHz results in a significant series resistance effect, with a decrease in the C_ox_. This effect is so strong that no accurate extraction of D_it_ and EOT is possible using conventional methodologies (high–low-frequency C-V method, conductance method, etc.) [22]. Secondly, a large positive shift of the C-V signal (and so of the V_th_) is seen while increasing the frequency. To our knowledge, no other groups have previously reported this Vth dependence on frequency in vertical GaN MOSFETs; however, similar phenomena have been observed on lateral fully recessed MOS-HEMT [23] and planar MOS capacitors [24]. The mechanisms are explained in more detail below.

In addition, the total C-V signal at 10 kHz seems divided into two parts: a first plateau between −2 V and +2 V, and the beginning of a second hump from 2 V to 8 V, in the case of the single-trench device (Figure 7a). A similar phenomenon with a different voltage range is observed for the multi-trench device (Figure 7b). This behavior differs from the signal at 200 Hz, where only one typical C-V characteristic is observed related to the device inversion, as observed for the single-trench device. Given the frequency dependence, the first C-V plateau could be the consequence of defects at the p-GaN MOS interface or at its vicinity [20], or even related to the p-GaN inversion [25]. In order to explore this assumption, we carried out TCAD C-V simulations at different frequencies by including positive fixed charge and donor defect densities at the p-GaN/dielectric interface in our vertical MOSFET.

The Structure Device Editor (SDE) module was used to design the simulated structure based on its expected thicknesses and doping, as well as the trench morphology illustrated in the TEM image in Figure 2. Electrical simulations were performed with the Synopsys S-Device module. The 2D model used for C-V simulations is based on a quasi-stationary AC model without considering the polarization charges intrinsic to the GaN/Al_2_O_3_ interface. To simplify the comparison with experimental results, the normalization is defined by the trench active area. Figure 8a depicts the simulated gate-to-source C-V characteristics at 10 kHz with and without defects (fixed charges and interface traps) at the p-GaN/dielectric interface. Two simulation cases are compared: case (A) includes only a high positive fixed charge density (+Q_fix_) while case (B) includes donor-type interface traps (n_t_) in addition to case A.

For the case (A), increasing +Q_fix_ from 1 × 10^12^ cm^−2^ to 7 × 10^12^ cm^−2^ strongly shifts the V_th_ toward negative values, which facilitates the inversion mode. In addition, this shift becomes so strong for high +Q_fix_ values (≥5 × 10^12^ cm^−2^) that it separates the capacitance contribution related to the inversion (sidewall p-GaN region) from that linked to the accumulation (trench bottom). This is confirmed in Figure 9 by the simulated electron density profiles located around the trench area, in the case of a +Q_fix_ value of 5 × 10^12^ cm^−2^ and for different device regimes (depletion, p-GaN inversion, accumulation).

For case B, we add a donor defect density at the p-GaN/dielectric interface, with a +Q_fix_ of 5 × 10^12^ cm^−2^. In this study, and based on the literature related to D_it_ for p-GaN MOS devices, the energy level chosen for n_t_ has been set at 0.3 eV from the valence band [20,26]. A clear parasitic capacitance appears in the depletion region which becomes wider while increasing the trap density from 1 × 10^11^ cm^−2^ to 7 × 10^12^ cm^−2^. However, a notable difference is observed compared to case A, especially within a voltage range of 1 to 6 V, where de-trapping phenomenon is the cause of the change in C-V characteristics. The second part of the curve between 6 V and 8 V (inversion/accumulation regime) seems unaffected by the trap concentration, but more related to the electrical effects brought by the high +Q_fix_ at the interface.

The effects of frequency on the C-V signal in case B are studied in Figure 10a. Frequencies ranging from 200 Hz to 10 kHz do not have any impact on the second part of the curve but the defects at the interface become more sensitive at lower frequency, allowing these types of trapped defects to be probed more precisely.

Comparing the physical effects observed between experimental C-V measurements (Figure 7) and simulated C-V (Figure 10) shows some similarities for device V_th_ value. However, a clear difference is observed on the impact of the frequency on the C-V signal. The simulated C-V characteristics at fixed frequency in case A demonstrate that adding a high fixed positive charge density at the p-GaN interface could shift V_th_ toward negative values while revealing an inversion signal occurring before the accumulation regime. In this case and considering a single-trench MOSFET device, the resulting simulated V_th, CV_ of ~2 V for a +Q_fix_ of 5 × 10^12^ cm^−2^ in Figure 8a is not far from the V_th, CV_ of 0.9 V extracted experimentally in Figure 7. This V_th_ difference compared to the simulated ideal Vth (~7 V) is still debated in the literature, but could be related to charge compensation linked to the GaN polarization charge, to an incomplete p-GaN doping activation [27,28], and/or to the integration of Al_2_O_3_ as gate dielectric [29,30].

The simulated C-V characteristics illustrated in Figure 10 in case B show that adding a high density of donor-type defects can bring a parasitic capacitance particularly sensitive to frequency in the depletion regime. In this case, the experimental C-V results (Figure 7) indicate two features that differ from the simulation, which could lead to the assumption that other physical effects must be taken into account in TCAD simulations for MOS devices.

The first feature is a capacitance plateau seen at 10 kHz, which could be related to defects at the p-GaN/dielectric interface. However, while this capacitance signal should be more pronounced when lowering the frequency, it unexpectedly disappears at 200 Hz for both devices. The second feature is a clear V_th_ positive shift that can be noticed when increasing the frequency, which is not observed for the simulated C-V characteristics. These two electrical phenomena could be related to the presence of a significant RC effect located at the trench bottom corners (Figure 2) [20,23]. This RC effect could be closely linked to the gate morphology, such as: (1) a high resistance related to a low electron density around the micro-trenching areas (Figure 9); (2) a high roughness on the trench sidewalls and trench bottom corners due to the trench-etching process; (3) a pronounced positive fixed charge trapping concentration around the trench bottom micro-trenching areas, possibly amplified by the high electric field around that region.

These three features could lead to a reduced mobility, especially around the trench bottom corners, which would affect the device response to the AC signal during C-V measurements. At low frequency (e.g., 200 Hz), the charges around the micro-trenching areas could respond to the AC signal due to the near quasi-static regime, and so only the capacitance of the whole gate (including trench bottom and trench sidewall signals) would be seen, as observed in Figure 7a. When increasing the frequency (e.g., 10 kHz), the charges around the micro-trenching could respond less to the increasing speed of AC signal, leading to a deformation of the C-V shape with a V_th_ shift, and so, to a loss of the trench bottom capacitance signal. Finally, at higher frequency (e.g., >128 kHz), the charges in that region could not respond to the fast AC signal, which means that the trench bottom C-V contribution would be filtered out with the increase in the frequency, resulting in the observation of a total C-V signal containing only the capacitance of the p-GaN channel and n^+^-GaN access region.

As a consequence, the first capacitance plateau observed at higher frequency could also give hints toward the signal of the p-GaN inversion. This would mean that, by taking advantage of a significant RC effect coming from the gate morphology, different frequencies could be used to separate the capacitance signal of the trench sidewalls from that of the whole device gate. Nevertheless, this RC effect around the gate remains a device issue that has a detrimental effect on the device electrical performances, as it increases the total device resistance and alters the device switching. Further studies will be conducted to reduce or suppress it, especially by improving the GaN trench etching and GaN/dielectric interface quality.

## 4. Conclusions

In conclusion, this work aimed to evaluate the transistor behavior of single- and multi-trench GaN-on-Si pseudo-vertical MOSFETs fabricated using a gate-first process flow based on an Al_2_O_3_/TiN MOS stack. The transistor devices showed normally OFF behavior, promising switching performances, and very low gate leakage current. Due to the extraction of a low effective channel mobility, we investigated the frequency-dependence of gate-to-source C-V characteristics of these devices. This analysis was also supported by TCAD simulations, where we studied the effects on the capacitance signal of adding positive fixed charge and donor defect densities at the p-GaN/dielectric interface, as well as the device response to frequency variations. Finally, by comparing our experimental and simulation results, we proposed a possible mechanism to explain the dependence of the V_th_ shift with the frequency, which could be influenced by a significant RC effect coming from the micro-trenching at the sharp trench bottom corners. This mechanism could also help in understanding the unexpected C-V features at high frequency, possibly linked to charge trapping at the p-GaN/dielectric interface, or more related to the p-GaN inversion of charge.

## Figures and Tables

**Figure 1 micromachines-16-01193-f001:**
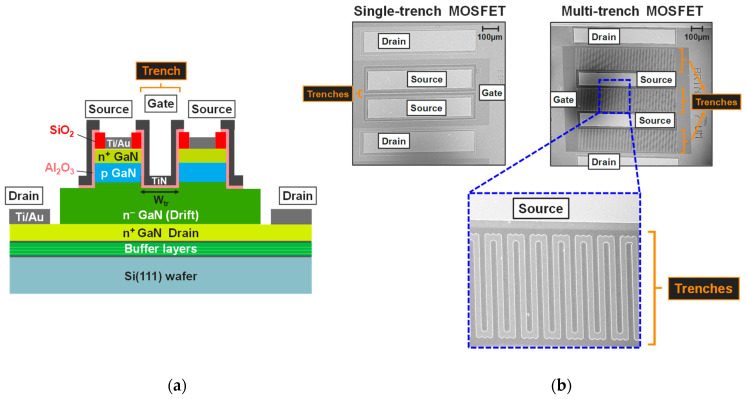
(**a**) Cross-sectional schematic of a single-trench pseudo-vertical GaN-on-Si MOSFET. (**b**) Top-view SEM images of single-trench and multi-trench devices.

**Figure 2 micromachines-16-01193-f002:**
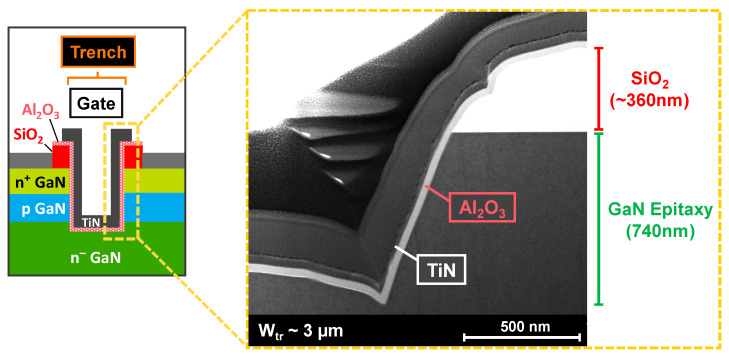
Bright-field-STEM image (**right**) of the right-hand trench sidewall of a single-trench pseudo-vertical MOSFET, as illustrated on the cross-sectional schematic (**left**).

**Figure 3 micromachines-16-01193-f003:**
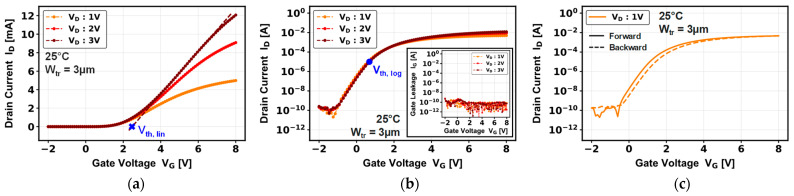
(**a**) Linear and (**b**) semi-log transfer characteristics in forward gate voltage with gate leakage current; (**c**) semi-log transfer characteristics in forward–backward gate voltage sweep, at V_D_ = 1 V, of a pseudo-vertical GaN-on-Si single-trench MOSFET.

**Figure 4 micromachines-16-01193-f004:**
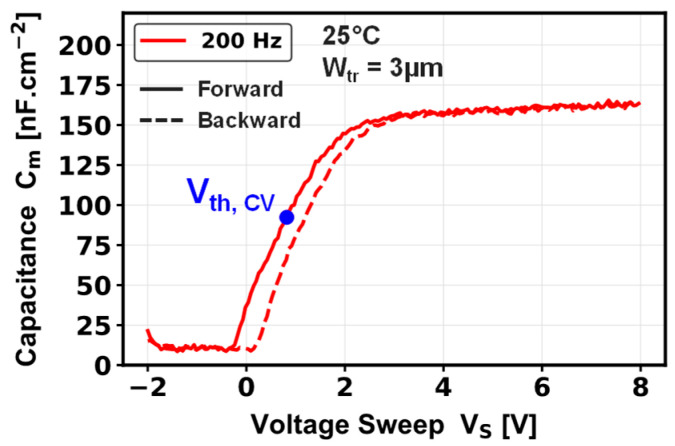
Gate-to-source capacitance–voltage characteristic, measured at 200 Hz, for a pseudo-vertical GaN-on-Si single-trench MOSFET.

**Figure 5 micromachines-16-01193-f005:**
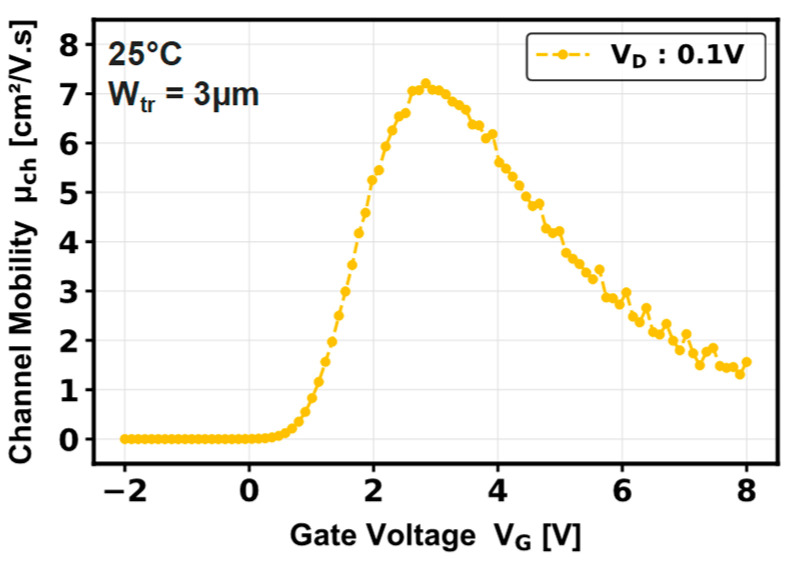
Field-effect channel mobility at V_D_ = 0.1 V of a pseudo-vertical GaN-on-Si single-trench MOSFET.

**Figure 6 micromachines-16-01193-f006:**
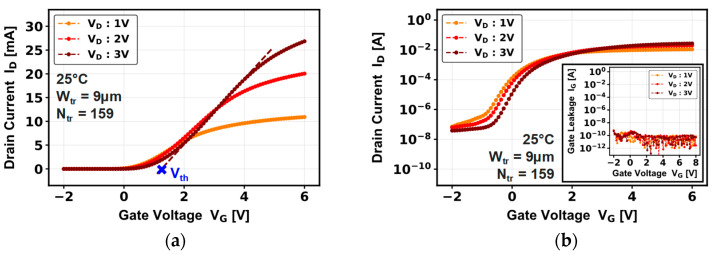
(**a**) Linear and (**b**) semi-log transfer characteristics with gate leakage current versus applied gate voltage, for a pseudo-vertical GaN-on-Si multi-trench MOSFET.

**Figure 7 micromachines-16-01193-f007:**
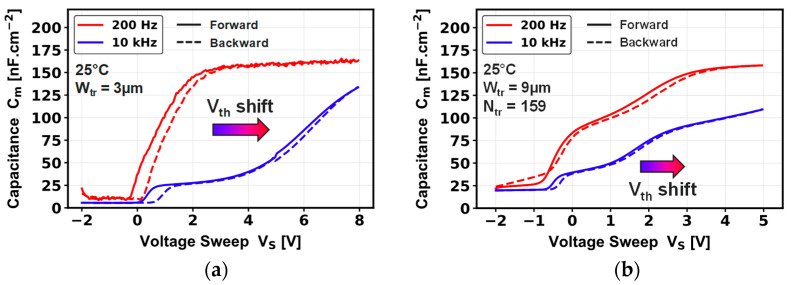
Gate-to-source capacitance–voltage characteristics, measured at 200 Hz and 10 kHz, for a pseudo-vertical GaN-on-Si (**a**) single-trench MOSFET, (**b**) multi-trench MOSFET.

**Figure 8 micromachines-16-01193-f008:**
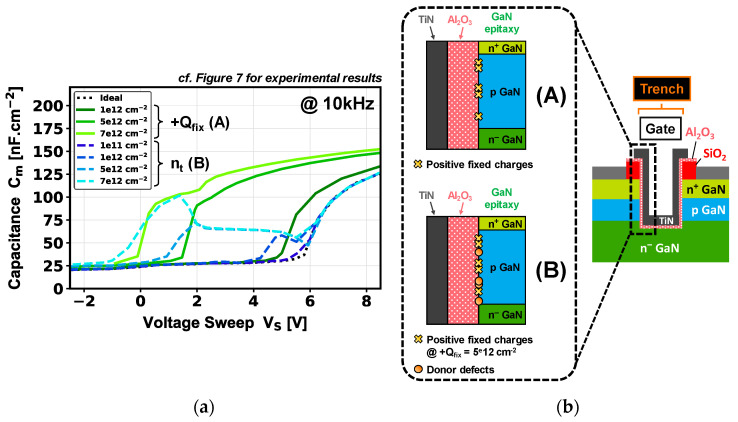
(**a**) Gate-to-source capacitance–voltage characteristics, simulated at 10 kHz, at various fixed charge densities (+Q_fix_, green curves) and donor defect densities (n_t_, blue curves) at the p-GaN/dielectric interface, for a pseudo-vertical GaN-on-Si single-trench MOSFET. (**b**) Cross-sectional schematic of the device MOS gate in the case of a positive fixed charge density (A) and a density of positive fixed charge and donor defects (B) added at the p-GaN/dielectric interface.

**Figure 9 micromachines-16-01193-f009:**
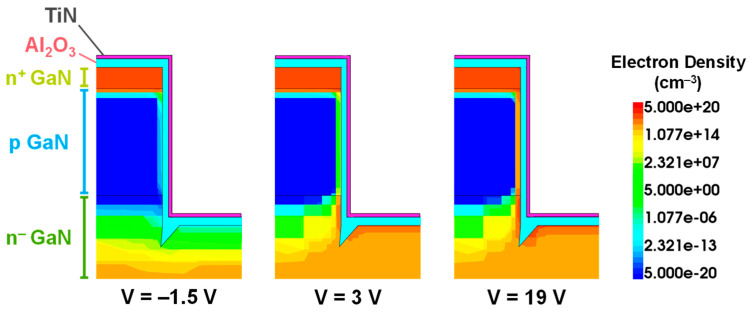
Simulated electron density around the trench area at different gate voltages, in the case of a +Q_fix_ of $5\times 10^{12}$ cm^−2^ at the p-GaN/dielectric interface, for a pseudo-vertical GaN MOSFET.

**Figure 10 micromachines-16-01193-f010:**
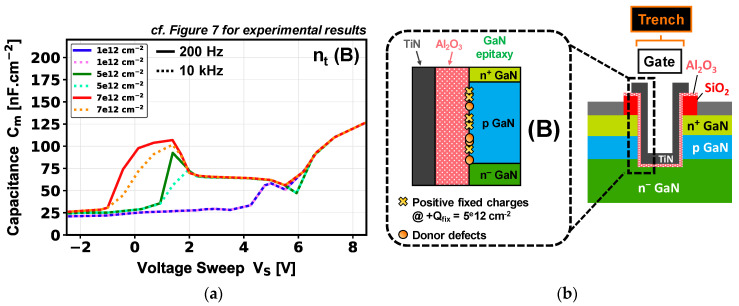
(**a**) Gate-to-source capacitance–voltage characteristics simulated at different frequencies, and as a function of donor-type densities at the p-GaN MOS interface. (**b**) Cross-sectional schematic of the device MOS gate in the case of a density of positive fixed charge and donor defects added at the dielectric/p-GaN interface (case B).

## Data Availability

The original contributions presented in the study are included in the article, further inquiries can be directed to the corresponding authors.

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
