# Peer review of "Electrical Characterization and Simulation of GaN-on-Si Pseudo-Vertical MOSFETs with Frequency-Dependent Gate C–V Investigation"

_micromachines, 2025, doi:10.3390/mi16111193_

Round 1

Reviewer 1 Report

Comments and Suggestions for Authors

The paper is about a pseudo-vertical MOSFET fabricated with GaN-on-Si with a gate first process. The device fabrication is well described and measurement results are clear. The low mobility triggered a gate C-V investigation leading to an evident frequency dependency. This behavior has been investigated with TCAD Sentaurus Synopsis simulations allowing to identify a possible mechanism leading to the Vth frequency dependent shifting: a significant RC effect that is due to the gate morphology. Presented are convincing and interesting. The paper is well written and publication worthy.

Author Response

Comment 1 :  The paper is about a pseudo-vertical MOSFET fabricated with GaN-on-Si with a gate first process. The device fabrication is well described and measurement results are clear. The low mobility triggered a gate C-V investigation leading to an evident frequency dependency. This behavior has been investigated with TCAD Sentaurus Synopsis simulations allowing to identify a possible mechanism leading to the Vth frequency dependent shifting: a significant RC effect that is due to the gate morphology. Presented are convincing and interesting. The paper is well written and publication worthy.

Response 1 : Thank you very much for the review of this work.

Reviewer 2 Report

Comments and Suggestions for Authors

This work deals with the fabrication and testing of two sets of pseudo-vertical GaN-on-Si MOSFETs. The Authors compare them in terms of their static ID-VDS characteristics as well as their C-V curves.

Some concerns arise from the TCAD modeling part.

The Authors do not show any calibration of the TCAD model over the experimental data: it would be useful to assess the accuracy of their conclusions drawn from simulations that aim at explaining the shift of VTH, for instance.

Author Response

Comments 1 : This work deals with the fabrication and testing of two sets of pseudo-vertical GaN-on-Si MOSFETs. The Authors compare them in terms of their static ID-VDS characteristics as well as their C-V curves.

Some concerns arise from the TCAD modeling part.

The Authors do not show any calibration of the TCAD model over the experimental data: it would be useful to assess the accuracy of their conclusions drawn from simulations that aim at explaining the shift of VTH, for instance.

Response 1 : Thank yo for pointing this out. Regarding the device structure, the TCAD model has been calibrated with respect of expected layer thicknesses, material properties, and gate spécifications (gate width, micro-trenching depth...) of our device.
On the electrical device response, Fig. 8 shows already a first step on the TCAD calibration methodology. The fixed charge as-well-as the defect densities have been varied through a wide range of densities, and their features (order of magnitude, energy level...) have been confirmed by the literature. Consequently, the Vth and the Cmax have been calibrated with respect to our experimental and literature datas. However, in our case, a proper calibration of the C-V curves would requires to take into consideration a pronounced RC effect in our TCAD  model.

As perspectives, we plan to do further electrical characterization or our device to assess the Dit of the gate interface, as well as electrical characterization to identify its origin. This would refine as well our model regarding the defect densities features. Also developing a TCAD model that implemantes a RC effect around the trench would also be a way to improve the TCAD calibration model and confirm our experimental results.

Minor corrections :
-Minor adjustement, precisions and mistake corrections affecting the presentation of the results, their explanations and the understanding of the article have been applied.
-Minor improvements of the figures have also been applied.
